# Upregulated Autophagy in Calcific Aortic Valve Stenosis Confers Protection of Valvular Interstitial Cells

**DOI:** 10.3390/ijms20061486

**Published:** 2019-03-25

**Authors:** Miguel Carracedo, Oscar Persson, Peter Saliba-Gustafsson, Gonzalo Artiach, Ewa Ehrenborg, Per Eriksson, Anders Franco-Cereceda, Magnus Bäck

**Affiliations:** 1Department of Medicine, Karolinska Institutet, 171 77 Stockholm, Sweden; Miguel.carracedo@ki.se (M.C.); Oscar.persson@ki.se (O.P.); psalgus@stanford.edu (P.S.-G.); gonzalo.artiach@ki.se (G.A.); ewa.ehrenborg@ki.se (E.E.); per.eriksson@ki.se (P.E.); 2Department of Molecular Medicine and Surgery, Karolinska Institutet, 171 77 Stockholm, Sweden; Anders.Franco-cereceda@ki.se; 3Theme Heart and Vessels, Division of Valvular and Coronary Disease, Karolinska University Hospital, 171 77 Stockholm, Sweden

**Keywords:** autophagy, calcification, calcific aortic valve stenosis, valvular interstitial

## Abstract

Autophagy serves as a cell survival mechanism which becomes dysregulated under pathological conditions and aging. Aortic valve thickening and calcification causing left ventricular outflow obstruction is known as calcific aortic valve stenosis (CAVS). CAVS is a chronic and progressive disease which increases in incidence and severity with age. Currently, no medical treatment exists for CAVS, and the role of autophagy in the disease remains largely unexplored. To further understand the role of autophagy in the progression of CAVS, we analyzed expression of key autophagy genes in healthy, thickened, and calcified valve tissue from 55 patients, and compared them with nine patients without significant CAVS, undergoing surgery for aortic regurgitation (AR). This revealed a upregulation in autophagy exclusively in the calcified tissue of CAVS patients. This difference in autophagy between CAVS and AR was explored by LC3 lipidation in valvular interstitial cells (VICs), revealing an upregulation in autophagic flux in CAVS patients. Inhibition of autophagy by bafilomycin-A1 led to a decrease in VIC survival. Finally, treatment of VICs with high phosphate led to an increase in autophagic activity. In conclusion, our data suggests that autophagy is upregulated in the calcified tissue of CAVS, serving as a compensatory and pro-survival mechanism.

## 1. Introduction

Thickening and calcification of the aortic valve is known as aortic valve sclerosis, eventually causing left ventricular outflow obstruction, and then referred to as calcific aortic valve stenosis (CAVS) [1]. CAVS is characterized by inflammatory processes, extracellular matrix remodeling, and trans-differentiation of valvular interstitial cells (VICs), the most abundant cell type in the aortic valve [2]. Initial fibrosis and valve thickening are followed by a subsequent calcification of the valve [3]. Importantly, CAVS incidence and severity increase with age [4,5] along with several cardiovascular risk factors, e.g., obesity [6], smoking [7], and renal dysfunction [8]. When severe, the two-year mortality rate of CAVS is almost 50% [9]. While valve replacement or percutaneous intervention can reverse this dismal prognosis, these interventions are not without risk in an elderly, fragile CAVS population. Therefore, understanding the underlying molecular mechanisms behind increased susceptibility as the disease progresses is of uttermost importance in order to develop a medical treatment to halt or reverse the calcification process of the aortic valve. 

Autophagy is a highly conserved mechanism involved in the homeostatic recycling and catabolism of misfolded proteins and damaged organelles in lysosomes [10]. Autophagy also serves as a cell survival mechanism in different diseases, including atherosclerosis [11,12], and can be dysregulated during aging [13]. Recently, autophagy has been implicated in vascular calcification, specifically, as a mechanism safeguarding vascular smooth muscle cells (VSMCs) from hyperphosphatemia [14]. These findings may be extrapolated to valve calcification, since VIC autophagy has been shown to inhibit pro-osteogenic signaling [15]. Taken together, these findings suggest that autophagy serves as a protective mechanism in VICs. However, how autophagy is regulated throughout the disease in human heart valves, and its role in VIC survival, is unknown.

Based on the above, the aims of the present study were (1) to establish the expression of autophagy pathways across the continuum of CAVS in human valves and (2) to determine the role of autophagy in VICs on cell survival and response to phosphate.

## 2. Results

### 2.1. Autophagy and Lysosomal Pathways Are Altered in Human Aortic Valves and Differentially Regulated under Calcification between CAVS and AR

To determine the regulation of autophagy during the progression of aortic valve sclerosis, we analyzed the gene expression levels of key autophagy genes in healthy, thickened, and calcified tissue from 55 CAVS patients. We included aortic valves derived from nine patients with aortic sclerosis, but without significant CAVS, undergoing surgery for aortic regurgitation (AR) for comparison. *ULK1* and *MAP1LC3A* expression levels were significantly downregulated in the calcified tissue in CAVS as compared with healthy and thickened tissue, whereas *MAP1LC3B* remained unchanged. On the other hand, *BECN1*, *ATG* 3, 7, 5, and 12 were significantly upregulated in the calcified tissue of CAVS valves as compared with healthy and thickened tissue. In sharp contrast, in the tissue from AR valves, only *ATG5* was significantly upregulated between healthy and thickened tissue, and *ATG12* between healthy and calcified tissue (Figure 1).

Another key step in autophagy is the degradation of the autophagosome, which is dependent on lysosomal activity. Gene expression analysis of the transcription factor EB (*TFEB*) revealed no significant differences in CAVS nor AR. Interestingly, gene expression of the lysosomal associated membrane protein (*LAMP1*) was downregulated only in the calcified tissue of CAVS valves as compared with thickened tissue. On the other hand, gene expression analysis of cathepsins revealed a strong upregulation of *CTS B*, *V*, and *L* in the calcified tissue of CAVS valves. In contrast, *CTSD* gene expression was downregulated in the calcified tissue of CAVS (Figure 2).

To understand the differential expression in autophagy genes observed in the valve tissue, we isolated VICs from human valves. Firstly, we explored the autophagic flux by inhibiting autophagosome degradation with bafilomycin. VICs derived from CAVS patients had significantly higher autophagic flux as compared with AR as measured by LC3 lipidation after bafilomycin treatment (Figure 3A).

### 2.2. Autophagy Serves as a Survival Mechanism in VICs

To understand the role of autophagy in VIC survival, we pharmacologically inhibited autophagy with bafilomycin. VICs treated with bafilomycin presented a significantly reduced cell viability as compared with control cells (Figure 3B).

Finally, to understand why autophagy was mainly differentially regulated in the calcified tissue, we treated VICs with high phosphate, with or without prior inhibition of autophagy by bafilomycin. This experiment revealed that high phosphate significantly increased LC3-II levels compared with controls when autophagy recycling was inhibited with bafilomycin (Figure 3C), indicative of increased autophagy.

## 3. Discussion

In the present study, autophagy arises as a differentially regulated process under calcifying conditions, specifically in the valves of patients with CAVS. Functionally, autophagy appears as a VIC survival mechanism increased under pro-calcifying conditions such as high phosphate.

Apart from a previous study showing ubiquitin positive staining in close proximity to heavily calcified nodules in valves from CAVS patients [16], little is known on autophagy pathways during valve calcification in CAVS. Studying the gene expression profile of key genes involved in autophagy in different parts of the same human aortic valve represents a unique model of the disease continuum as the aortic valve progresses from healthy, to thickened, and then to calcified tissue. Using this model, we here identify that expression levels of autophagy pathways were upregulated exclusively in the calcified valve tissue of CAVS. In addition, in the calcified tissue we observed a downregulation in *ULK1*, a key ATG in the early stages of autophagy, and a general upregulation of the ATGs involved in the later stages of autophagy [17]. These observations suggest an upregulation of autophagy in the calcification process during CAVS progression, which is compensated by downregulating *ULK1*. In further support of increased autophagy in the continuum of valve calcification in CAVS, we observed an increase in gene expression of the lysosomal proteases CTS B, L, and V, essential in the execution of the autophagic flux [18]. Whereas cathepsins B, L, and D have not previously been explored in aortic valves, the elastolytic cathepsins S, K, and V are known to be upregulated in CAVS [19]. The lack of changes in expression levels of the transcription factor TFEB, which regulates these pathways [20], indicate that TFEB activity, rather than expression levels, may be increased in CAVS. Importantly, similar differences in the regulation of either the autophagy or lysosomal pathways were not observed in sclerotic valves without CAVS, used as comparison in the present study. Taken together, these results suggest that autophagy flux is increased exclusively in the calcified tissue of CAVS valves.

To test this hypothesis, we measured the autophagic flux in vitro in VICs derived from either CAVS or AR patients. These experiments revealed an increased LC3-II lipidation in VICs derived from CAVS patients as compared with those derived from sclerotic AR valves. These data hence suggest an increased autophagic activity, specifically in the autophagosome formation, between the more severe calcification in CAVS as compared with sclerotic AR valves, and that these processes are active in VICs. Interestingly, VSMC autophagy has been proposed as a protective mechanism against vascular calcification, being upregulated in the arteries of both hyperphosphatemic rats and BBA/2 mice fed a high phosphate diet [14,21]. Of importance, inhibition of autophagy with 3-methyadenine (3-MA) or by knocking down Atg5 led to an increase in vascular calcification, whereas induction of autophagy with rapamycin or valproic acid inhibited in vitro calcification. In line with this data, inhibition of autophagy by bafilomycin and knock down of Atg7 in VICs resulted in an increase of both calcification and of the pro-osteogenic molecules bone morphogenic protein 2 (BMP-2) and alkaline phosphatase (ALP). Additionally, induction of autophagy in VICs by rapamycin led to a decrease of both ALP and BMP-2; however, whether this translated to a reduction of calcification in VICs was not explored [15].

In the preset study, inhibition of the degradation step of autophagy by bafilomycin A1 led to a decrease in VIC viability, which could explain the increase in calcification observed by Deng et al. [15], especially since a decrease in autophagy may increase VIC cell death, thus serving as nucleating structures for calcification [22]. Moreover, we show here that acute high phosphate treatment increased autophagic flux in VICs when the recycling of the autophagosome was inhibited by bafilomycin, implying that an upregulation in autophagy might precede VIC calcification.

The results of the present study provide a comprehensive overview of autophagy pathways through different stages of CAVS and indicate a functional role of autophagy in VICs. Nonetheless, certain limitations should be acknowledged. The study samples for CAVS and AR were unequal which is why we refrained from direct comparisons between the two groups. However, the differences detected between healthy and diseased tissue in CAVS, but not in AR, points to a differential regulation of autophagy-induced gene expression in CAVS and AR. The impact of autophagy inhibition in VIC viability and calcification should be further confirmed with other approaches such as pharmacological inhibition by, for example, 3-methyladenine or ATG knock-down experiments, to both determine the key stage of autophagy in cell viability and to rule out possible direct effects of bafilomycin on apoptosis [23].

In summary, our data suggest that autophagy is upregulated in valves from CAVS patients. By using primary cultures of human aortic VICs, we suggest that the upregulation of autophagy observed in the calcified tissue of these valves serves as a compensatory and pro-survival mechanism to protect against calcification.

## 4. Materials and Methods

### 4.1. Patients

Aortic valves were obtained after valve replacement surgery from patients with CAVS (*n* = 55) and with AR (*n* = 9). All patients gave informed consent, and the study was approved by the local ethics committee (October 24, 2012/1633).

### 4.2. Sample Preparation and Macroscopic Dissection

Explanted valves were placed in RNAlater (Qiagen) at 4 °C right after surgery. Valves were then macroscopically dissected into healthy, thickened, or calcified tissue, as described previously [24,25], and frozen at −80 °C until RNA extraction.

### 4.3. RNA Extraction and Quality Assessment

Total RNA from valves was isolated using the RNeasy Lipid Tissue Mini kit (Qiagen, Hilden, Germany). Quantification and quality of RNA was assessed utilizing a NanoDrop (Thermo Scientific, Waltham, MA, USA) and a 2100 Bioanalyzer (Agilent, Santa Clara, CA, USA), respectively. Valve gene expression data was obtained using Gene Chip Affymetrix human transcriptome 2.0 (HTA 2.0 arrays, Santa Clara, CA, USA) and normalized with Signal space transformation-robust multi-chip analysis (SST-RMA) using Expression Console (Affymetrix, Santa Clara, CA, USA).

### 4.4. VIC Isolation, Culture, Treatment, and Survival

Valves utilized for primary cell cultures obtained from valve replacement surgery were conserved at 4 °C in cell culture medium (DMEM, 10% FBS, 100 U/mL penicillin, 100 ug/mL streptomycin, 1mM sodium pyruvate, 10mM HEPES and 2 mM glutamine) (Gibco, Waltham, MA, USA) until the 16 h enzymatic digestion with collagenase I and dispase II (Sigma, St. Louis, MO, USA). VICs were then seeded in polystyrene plates and the culture medium exchanged every two days. VICs were used between passages 1 and 3.

For the assessment of autophagic flux, VICs were stimulated for 2 h with bafilomycin A1 or vehicle control (DMSO) and thoroughly washed with PBS before further treatments. We prepared 2.6 mM phosphate as previously described [26] and added it to the VICs for 24 h.

Proliferation was assessed by WST-1 reagent (Roche) according to manufacturer’s protocol after treatment with bafilomycin A1 or vehicle control (DMSO) for 24 h.

### 4.5. Immunoblotting

For microtubule-associated protein 1A/1B-light chain 3 (LC-3), immunoblotting protein extracts were loaded on a 14% acrylamide SDS-PAGE gels and transferred to PVDF membranes (BioRad, Hercules, CA, USA). The membranes were blocked using 5% milk and incubated with primary antibodies against LC-3B (Novus Biologicals, Centennial, CO, USA, NB100-2220). A HRP-conjugated secondary antibody (BioRad) was used to amplify the signal. Blots were developed using enhanced chemiluminescence reagent kit (GE Healthcare, Chicago, IL, USA). ImageJ was used for densitometry, and autophagy flux was defined as LC3-II fold-change between experimental conditions supplemented with bafilomycin A1 over experimental condition without bafilomycin A1 by using LC-II density normalized to β-actin.

### 4.6. Statistical Analyses

Results are expressed as mean ± S.D. Statistical significance of differences for normally-distributed data was assessed with one-way ANOVA followed by recommended post hoc tests for multiple comparisons. For non-normally distributed data, statistical significance was assessed using the Wilcoxon signed test when comparing two paired groups, the Mann-Witney test when comparing two unpaired groups, and with a Kruskal-Wallis test followed by Dunn’s test for multiple comparisons. Statistical significance was assigned at *p* < 0.05. Statistical analyses were performed using GraphPad Prism 7 (GraphPad Software Inc., La Jolla, CA, USA).

## Figures and Tables

**Figure 1 ijms-20-01486-f001:**
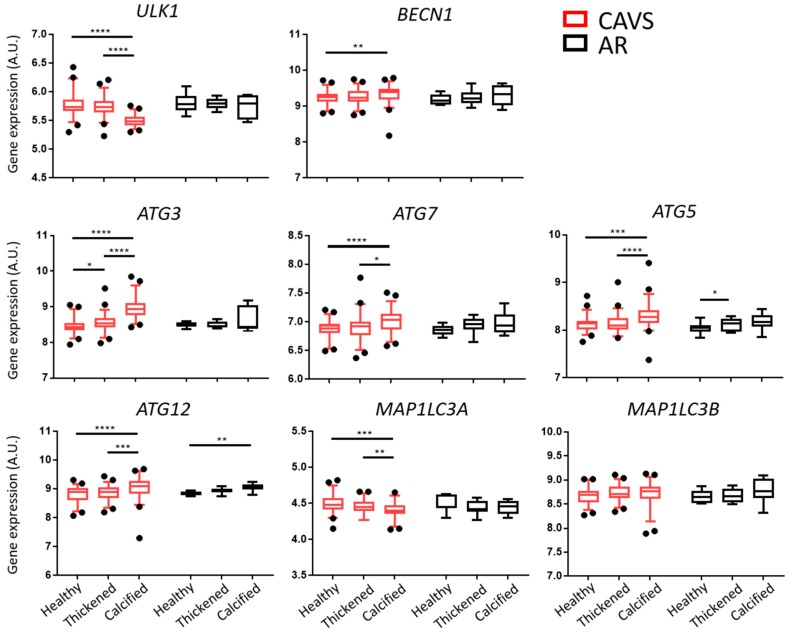
Autophagy pathways are differentially regulated in the calcified tissue of human valves. Gene expression analyses of key autophagy genes in the healthy, thickened, and calcified tissue of human valves from calcific aortic valve stenosis (CAVS; *n* = 55) and aortic regurgitation (AR; *n* = 9) patients. Data presented as box-and-whisker plots with 5–95 percentiles. * = *p* < 0.05, ** = *p* < 0.01, *** = *p* < 0.001, **** = *p* < 0.0001.

**Figure 2 ijms-20-01486-f002:**
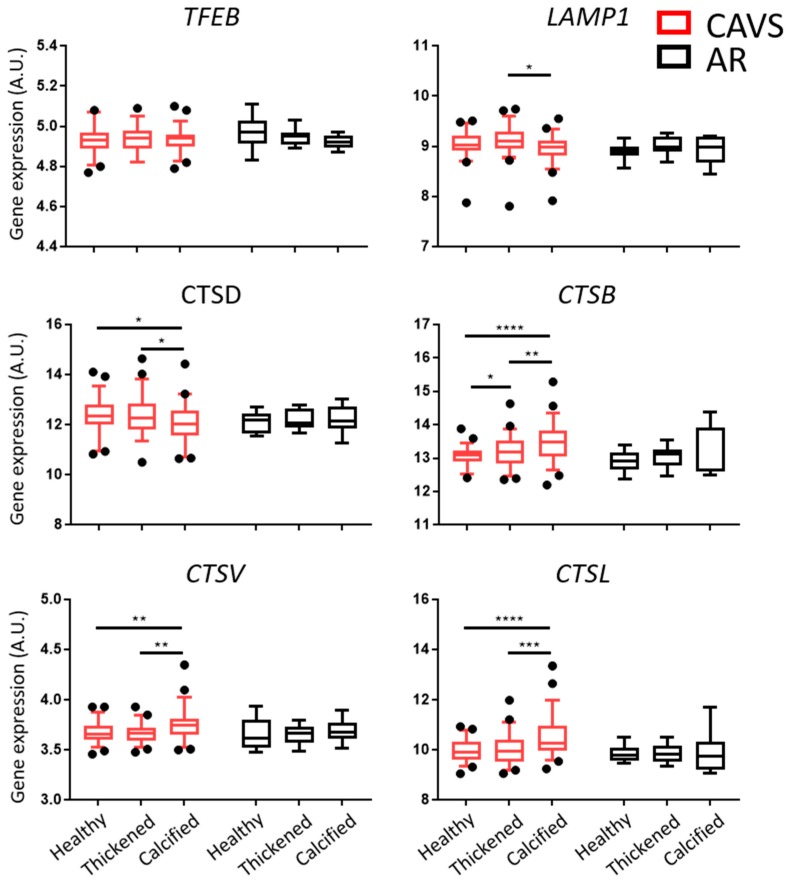
Lysosomal pathways are differentially regulated in the calcified tissue of human valves. Gene expression of key lysosomal genes in the healthy, thickened, and calcified tissue of human valves from calcific aortic valve stenosis (CAVS; *n* = 55) and aortic regurgitation (AR; *n* = 9) patients. Data presented as box-and-whisker plots with 5–95 percentiles. * = *p* < 0.05, ** = *p* < 0.01, *** = *p* < 0.001, **** = *p* < 0.0001.

**Figure 3 ijms-20-01486-f003:**
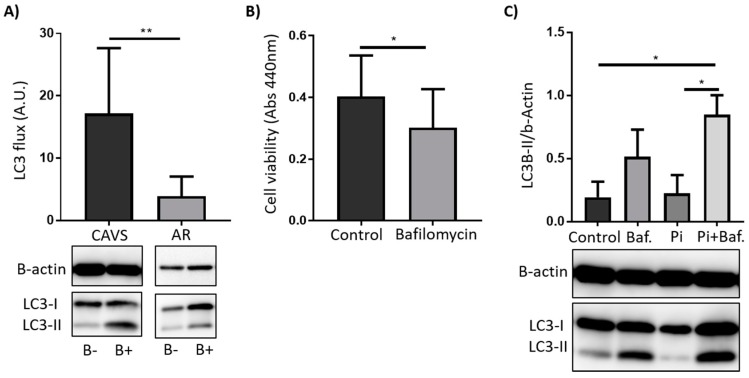
Autophagic activity is differentially expressed in AR and in CAVS and serves as a survival mechanism in valvular interstitial cells (VICs). (**A**) Immunoblotting against LC3-I and LC3-II in human VICs treated for 2 h with bafilomycin (1nM) (B+) or vehicle control (DMSO) (B-), including quantification of the autophagic flux measured as the LC3-II ratio between (B+/B-)/β-actin. CAVS: *n* = 5, AR: *n* = 4. (**B**) VIC viability from CAVS patients after 24 h treatment with bafilomycin (1nM) or vehicle control (DMSO) *n* = 7. (**C**) Immunoblotting against LC3-I and LC3-II in human VICs from CAVS patients treated for 2 h with bafilomycin (1nM) or vehicle control (DMSO) followed by 24 h treatment with 2.6 mM phosphate (Pi) or control (media) *n* = 4. Quantified as the densiometry of LC3-II/β-actin. Data presented as mean ± S.D. * = *p* < 0.05, ** = *p* <0.01, *** = *p* < 0.001, **** = *p* < 0.0001.

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
