# Peer review of "Upregulated Autophagy in Calcific Aortic Valve Stenosis Confers Protection of Valvular Interstitial Cells"

_ijms, 2019, doi:10.3390/ijms20061486_

Round 1

Reviewer 1 Report

This paper explores the role of autophagy in CAVS. Taking into account that the authors worked with patient samples, I understand is difficult to get enough samples. There is a big gap between the number of samples from CAVS patients (55) vs. the number of healthy patients (9). I know the statistics retrieves a significant difference in terms of expression between these two groups, but based on the number of samples I was wondering if this is realistic.

Respect to the second part, the authors should clearly explain that the VICS were isolated from both CAVS and AR patients. Additionally, bafilomycin and Phosphate, are stimulators or inhibitors of autophagy? It is not clear in the text. Why don’t you analyze in this part the same autophagy genes as in the first part? 

There are two figures 1. In the second figure 1, please clearly state in b) and c) parts that you are talking about CAVS derived VICS.

Author Response

REVIEWER 1

This paper explores the role of autophagy in CAVS. Taking into account that the authors worked with patient samples, I understand is difficult to get enough samples.

1_There is a big gap between the number of samples from CAVS patients (55) vs. the number of healthy patients (9). I know the statistics retrieves a significant difference in terms of expression between these two groups, but based on the number of samples I was wondering if this is realistic.

We acknowledge that the CAVS and AR patient samples are unequal and that group comparisons should be interpreted with care. For that reason, we have in the revised manuscript taken away the direct comparisons between CAVS an AR and have now re-analyzed the two groups separately. These results, shown in the revised Figure 1, show the differential expression of autophagy-related genes in valves derived from CAVS but not AR patients. We also acknowledge the sample size difference as a limitation in the revised manuscript (Discussion, Page 7; lines 179-180).

2_Respect to the second part, the authors should clearly explain that the VICS were isolated from both CAVS and AR patients.

Since Fig 2A in the original submission (Fig 3A in the revised manuscript) showed differences between CAVS and AR, we agree that using VICs from both CAVS and AR patients in subsequent experiments was not optimal. In the revised manuscript, we have clarified in the figure legend that VICs were isolated from both CAVS and AR patients for the experiments presented in Fig 3A in the revised manuscript. Furthermore, we have taken away the results obtained with AR VICs so that panels B and C now only contain results obtained in VICs from CAVS patients.

3_Additionally, bafilomycin and Phosphate, are stimulators or inhibitors of autophagy? It is not clear in the text.

We thank the reviewer for the comment. We have now re-written the results (page 5-6, lines 109-114) and the discussion (page 8, lines 174-177) to clarify that phosphate acts as an inducer of calcification whereas bafilomycin is an inhibitor of the recycling of the autophagosome.

4_Why don’t you analyze in this part the same autophagy genes as in the first part?

While the extensive exploration of gene expression is whole valves provided observational data on the associations between key autophagy pathways and different stages of disease in CAVS and AR, we focused on functional readouts in VICs in the second part to show direct effects on autophagic flux and cell viability. The point is well taken, and we agree that further explorations in VICs using for example knock-down experiments of the autophagy genes would help to decipher the exact mechanisms of VIC autophagy (see also response to Reviewer 1, Comment 1). This is now indicated in the new limitations section of the revised manuscript (Page 7; lines 179-180).

5_There are two figures 1. In the second figure 1, please clearly state in b) and c) parts that you are talking about CAVS derived VICS.

Please see response to point 2

REVIEWER 2

The manuscript ID ijms-464804 entitled, "Dysregulated autophagy in calcific aortic valve stenosis" is aiming to evaluate the autophagy activity and to elucidate the role of autophagy in the calcific aortic valve stenosis (CAVS) patients. The study aim is clear and significant. Some data are intriguing. However, the experimental design, the data interpretations and the entire conclusion are ambiguous and really confusing. Even in the abstract,

1. The authors claimed that they used bafilomycin to inhibit autophagy but also it activated autophagy with high-phosphate.

2. The authors argued the “dysregulation in autophagy” and also autophagy is upregulated for protection.

Entire revision of concept is strongly suggested.

We thank the reviewer for these comments. As suggested, we have clarified the experimental design and data interpretation, and revised the concept on dysregulation. We have addressed each of the comments raised, as detailed below.

Major:

1. The interpretation of Figure2 seems misleading.

Bafilomycin A1 inhibits the degradation step of autophagy (1-3). It is used for autophagy flux assay. Based on the concept of autophagy flux assay, the LC3-II accumulation by baf treatment in the Fig2C may simply correspond to the inhibition of LC3 degradation. Fig2C result suggests that high-phosphate per se upregulates autophagy especially at the lysosomal degradation step. The impact of autophagy inhibition should be confirmed with multiple approaches, such as 3-methyladenine, or knocking-down of ATG genes.

We thank the reviewer for the comment. We have now modified the interpretation of the data as the reviewer suggests. This is now changed in the results section (page 5-6, lines 109-114), as well as in the discussion (page 7, lines 172-174). The increase in the transcription of the cathepsins would argue in favor of an increase in the lysosomal flux of CAVS patients over AR patients.

We acknowledge that to fully understand the stage in which autophagy is affected in CAVS we would require to inhibit it utilizing the aforementioned approaches. This has now been added as a limitation in the discussion (page 7-8, lines 182-186). Nonetheless, to our knowledge, we are the first to provide a comprehensive overview of the autophagy status, within the different stages of disease.

In addition, examining the gene expressions of lysosomal genes and the related regulator, such as LAMP1, Cathepsins, and TFEB, might help for providing more reasonable interpretation.

We thank the reviewer for raising this point. We have now explored the expression levels of the different proposed genes in human aortic valves. These results are shown in the new Fig 2 tissues studies and have updated results section (page 2, lines 76-82), and discussion (page 7, lines 142-151).

2. I don’t get the point of Fig2B experiment.

Although the concentration of bafilomycin A1 is relatively low, it has already been known that is toxic and enough to induce cell death via not only autophagy but also apoptosis pathway (4). The impact of autophagy on the cell viability should be examined under stress conditions mimicking the development of CAVS.

We thank the reviewer for the comment. However, we would like to emphasize that this cells are primary cultures obtained from calcified valves and used at low passage to retain the conditions of CAVS as closely as possible. This is illustrated by the differences in autophagy between CAVS and AR VICs, as shown in Fig 3A in the revised manuscript (Fig 2A in the original submission). As suggested, we have therefore in the revised manuscript included only VICs derived from CAVS patients in Panel B of the revised Figure to mimic the stress conditions during the development of CAVS. Nonetheless, we acknowledge that the effect of bafilomycin on apoptosis cannot be ruled out and this has now been added as a limitation in the discussion (page 8, line 185-186).

3. Figure legends and methods are too brief and thus it’s hard to follow what the data are exactly showing

We apologize for this. The current report was initially sent as a brief report. However, it has now been updated to a Communication format, and material and methods, and specially figure legends adjusted and explained in more detail.

In Figure 1, what are the black dots on the red chart? If they indicate data points, are they only 4 samples?

We apologize for not clarifying in the figure legend that the data is presented as box-and-whisker with 5-95 percentile. Therefore, the black dots represent outliers that are not contained within the whiskers of the box plot.

Figure 2 is really confusing. What is the Fig2A immunoblotting for? How did the authors measure the autophagy? What does the bar graph indicate?

We thank the reviewer for raising this point. In panel A of Fig 3 in the revised manuscript (Fig 2 in the original submission), autophagy is measured as the LC3-II ratio between the Bafilomycin A1 (B+) over the control (DMSO) (B-), using LC-II density, normalized to b-actin.

We would like to apologize for the confusion, we have now updated the figure legend to clarify it. And further clarified it in the results section (Page 2, lines 83-86)

Minor:

Fig2 The antibody used in this study must be anti LC3B but LC3A. Please confirm the catalog number.

We thank the reviewer for raising this point. The antibody is indeed against LC3B. We have now clarified this in the material and methods section page 9 lines 230-231. The catalog number is present in the material and methods section.

https://www.novusbio.com/products/lc3b-antibody_nb100-2220

Fig1 qRT-PCR targets LC3A. In general, LC3B is a standard autophagy marker rather than LC3A. The expression of MAP1LC3B should be shown in Fig1.

We have completed the data in Fig 1 with new data on MAP1LC3B as suggested.

Figure 2 legend is labeled “Figure 1”

We thank the reviewer for noticing this mistake. It has now been changed.

(References)

1. Klionsky DJ et al. Guidelines for the use and interpretation of assays for monitoring autophagy (3rd edition). Autophagy. 2016;12:1-222.

2. Zhang, X. J., et al. (2013). "Why should autophagic flux be assessed?" Acta Pharmacol Sin 34(5): 595-599.

3. Mizushima, N. and T. Yoshimori (2007). "How to interpret LC3 immunoblotting." Autophagy 3(6): 542-545.

4. Yuan N, Song L, Zhang S, Lin W, Cao Y, Xu F, Fang Y, Wang Z, Zhang H, Li X, Wang Z, Cai J, Wang J, Zhang Y, Mao X, Zhao W, Hu S, Chen S, Wang J. Bafilomycin A1 targets both autophagy and apoptosis pathways in pediatric B-cell acute lymphoblastic leukemia. Haematologica. 2015;100:345–356.

Reviewer 2 Report

The manuscript ID ijms-464804 entitled, "Dysregulated autophagy in calcific aortic valve stenosis" is aiming to evaluate the autophagy activity and to elucidate the role of autophagy in the calcific aortic valve stenosis (CAVS) patients. The study aim is clear and significant. Some data are intriguing. However, the experimental design, the data interpretations and the entire conclusion are ambiguous and really confusing. Even in the abstract,

1. The authors claimed that they used bafilomycin to inhibit autophagy but also it activated autophagy with high-phosphate.  

2. The authors argued the “dysregulation in autophagy” and also autophagy is upregulated for protection. 

Entire revision of concept is strongly suggested.

Major:
1. The interpretation of Figure2 seems misleading.
Bafilomycin A1 inhibits the degradation step of autophagy (1-3). It is used for autophagy flux assay. Based on the concept of autophagy flux assay, the LC3-II accumulation by baf treatment in the Fig2C may simply correspond to the inhibition of LC3 degradation. Fig2C result suggests that high-phosphate per se upregulates autophagy especially at the lysosomal degradation step. The impact of autophagy inhibition should be confirmed with multiple approaches, such as 3-methyladenine, or knocking-down of ATG genes.

In addition, examining the gene expressions of lysosomal genes and the related regulator, such as LAMP1, Cathepsins, and TFEB, might help for providing more reasonable interpretation.

2. I don’t get the point of Fig2B experiment.

Although the concentration of bafilomycin A1 is relatively low, it has already been known that is toxic and enough to induce cell death via not only autophagy but also apoptosis pathway (4). The impact of autophagy on the cell viability should be examined under stress conditions mimicking the development of CAVS.

3. Figure legends and methods are too brief and thus it’s hard to follow what the data are exactly showing

In Figure 1, what are the black dots on the red chart? If they indicate data points, are they only 4 samples?  

Figure 2 is really confusing. What is the Fig2A immunoblotting for? How did the authors measure the autophagy? What does the bar graph indicate?

Minor:
Fig2 The antibody used in this study must be anti LC3B but LC3A. Please confirm the catalog number.

Fig1 qRT-PCR targets LC3A. In general, LC3B is a standard autophagy marker rather than LC3A. The expression of MAP1LC3B should be shown in Fig1.

Figure 2 legend is labeled “Figure 1”

(References)
1. Klionsky DJ et al. Guidelines for the use and interpretation of assays for monitoring autophagy (3rd edition). Autophagy. 2016;12:1-222.
2. Zhang, X. J., et al. (2013). "Why should autophagic flux be assessed?" Acta Pharmacol Sin 34(5): 595-599.
3. Mizushima, N. and T. Yoshimori (2007). "How to interpret LC3 immunoblotting." Autophagy 3(6): 542-545.
4. Yuan N, Song L, Zhang S, Lin W, Cao Y, Xu F, Fang Y, Wang Z, Zhang H, Li X, Wang Z, Cai J, Wang J, Zhang Y, Mao X, Zhao W, Hu S, Chen S, Wang J. Bafilomycin A1 targets both autophagy and apoptosis pathways in pediatric B-cell acute lymphoblastic leukemia. Haematologica. 2015;100:345–356.

Author Response

(The authors gave the same response as above.)

Round 2

Reviewer 1 Report

Thank you for addressing all of my comments and concerns. Now the article is perfect in this form. 

Reviewer 2 Report

I appreciate the great efforts that the authors have made in response to my questions and concerns. The revision clarifies all the points I raised and is acceptable in the current form.

Please ensure if the Y title of Fig3C is correct.